# Solving Task Scheduling Problems in Dew Computing via Deep Reinforcement Learning

Pablo Sanabria [1,2], Tomás Felipe Tapia [1], Rodrigo Toro Icarte [1,2] and Andres Neyem [1,2,*]

1   Computer Science Department, Pontificia Universidad Catolica de Chile, Santiago 7820436, Chile; psanabria@uc.cl (P.S.); tetapia@uc.cl (T.F.T.); rtoro@uc.cl (R.T.I.)
2   Centro Nacional de Inteligencia Artificial CENIA, Santiago 7820436, Chile
*   Correspondence: aneyem@uc.cl

**Abstract:** Due to mobile and IoT devices' ubiquity and their ever-growing processing potential, Dew computing environments have been emerging topics for researchers. These environments allow resource-constrained devices to contribute computing power to others in a local network. One major challenge in these environments is task scheduling: that is, how to distribute jobs across devices available in the network. In this paper, we propose to distribute jobs in Dew environments using artificial intelligence (AI). Specifically, we show that an AI agent, known as Proximal Policy Optimization (PPO), can learn to distribute jobs in a simulated Dew environment better than existing methods—even when tested over job sequences that are five times longer than the sequences used during the training. We found that using our technique, we can gain up to 77% in performance compared with using human-designed heuristics.

**Keywords:** Dew computing; reinforcement learning; scheduling algorithms





## 1. Introduction

The massive growth of computation-intensive tasks in mobile applications has been imposing heavy computation demands on resource-constrained devices in recent years. Unfortunately, devices such as smartphones or IoT devices cannot usually meet those demands. To tackle this problem, *Dew computing* proposes to *offload* computation-intensive jobs to more powerful (nearby) devices [1–5]. A nearby device is a device that is connected to the same local network. The idea is to, for instance, send jobs from your smartphone to your laptop so each job is executed faster and does not drain your smartphone's battery. For Dew computing to work well in practice, however, the key challenge is to understand how to effectively distribute jobs across nearby devices.

This work addresses the problem of task scheduling in Dew environments. A Dew environment consists of a set of devices connected to a local network. The devices might vary in their features and capabilities, including their power source (and battery capacity), number of processor/CPU cores, storage, and sensors. In addition, users might interact with some of these devices at different times. All these factors must be taken into account to effectively distribute tasks in a Dew environment.

To distribute tasks in Dew environments, current methods follow human-designed policies. These policies try to balance the workload among the devices by following a set of predefined rules. Some examples include the *Simple Energy-Aware Scheduler (SEAS)* [6], the *Batch Processing Algorithm (BPA)* [7], and *Round Robin (RR)* [8]. Unfortunately, these methods are unable to adapt to the particular features of a given Dew environment and, as a result, they make suboptimal decisions and waste valuable resources.

In this paper, we propose to learn how to distribute jobs in a Dew environment using reinforcement learning (RL) [9]. RL is a subfield of artificial intelligence that studies how to develop agents that can learn optimal behavior by interacting with an environment. Every interaction with the environment delivers a reward signal that the agent seeks to maximize.

To accomplish this, the agent improves its current policy (a mapping from observations to actions) by learning from its past experiences. Powered by deep learning, RL agents have been used to solve complex decision-making problems across different research areas, from robotics [10] to conversational agents [11] and drug discovery [12]. Here, we propose to let an RL agent learn how to effectively distribute jobs in a Dew environment.

Previous works have explored the use of RL for distributing jobs in Edge and Cloud computing [13–16]. However, RL has not been tried in Dew computing. Dew computing has its own particular challenges that do not normally arise in Cloud computing or Edge computing. For instance, in Dew computing, some of the devices might run out of battery or users might start interacting with them. In addition, existing works have not studied whether RL agents can learn policies that generalize well to new situations. Our work shows that deep RL agents can learn to offload tasks better than state-of-the-art heuristic methods, even when tested in previously unseen situations.

The main contributions of our work are then as follows. First, we propose to use Deep RL to distribute jobs in a Dew environment. By Deep RL, we refer to a family of RL agents that use deep neural networks to encode policies. In contrast to tabular RL (such as Q-learning [17] or SARSA [18]), Deep RL algorithms can learn policies that generalize to unseen situations and solve problems with continuous state spaces—as is the case in Dew computing. Specifically, we used a Deep RL agent, called *Proximal Policy Optimization (PPO)* [19], to offload tasks in a *fixed* Dew environment (i.e., a Dew environment with a fixed configuration of devices). By following a trial-and-error strategy, the agent learns about the features of each device and how to assign jobs so that they are completed as soon as possible.

Our second contribution is to develop an interface that connects OpenAI Gym [20] with a state-of-the-art Dew computing simulator [21]. This interface allows us to quickly test different RL methods for task offloading in Dew environments. In fact, our interface allowed us to test another deep RL method, called A3C [22], but its performance was considerably worse than PPO.

Finally, we empirically demonstrate that by constantly showing new situations to the agent, the agent learns to generalize: that is, to appropriately distribute sequences of jobs that arrive in patterns and sizes that the agent has not seen during training.

As a brief summary of our empirical findings, we discovered that RL agents can learn to effectively distribute jobs in fixed Dew environments—largely outperforming state-of-the-art heuristics with respect to the number of instructions per second that are executed in the environment. To do so, however, the agent must be trained for a long time. As such, there is a trade-off. On the one hand, human-designed heuristic methods can be applied to any Dew environment and perform reasonably well right away. On the other hand, RL agents achieve impressive performance in the long term, but in the short term, they would not perform much better than a heuristic that randomly assigns jobs to devices. Whether it pays off to rely on RL agents over heuristic methods is application dependent.

## 2. Edge and Dew Computing

Mobile Edge Computing is a paradigm that seeks to solve latency and network traffic problems found in Mobile Cloud Computing environments [23]. Khan et al. [24] define Edge Computing as "*a model that allows a cloud-based computing capacity providing services making use of the infrastructure that is on the edge of the network.*" In this way, Mobile Edge Computing allows the use of servers or workstations that are within a computer network, thus ensuring low latency and at the same time enabling the efficient processing of information, which permits the deployment of more robust applications. Furthermore, this paradigm also lets Edge servers work with other nodes in proximity or collaborate with Cloud services, thus deploying much larger and more efficient applications [25,26]. However, while Edge computing helps to reduce the network's problems, it still depends on the network backbone that may not be available or reachable in certain situations (such as working with IoT devices in mines and on ships, in deserts, or moving vehicles).

Dew computing is a new paradigm where connected devices offload jobs to nearby devices in the same network. This paradigm proposes an architecture that tries to reduce network latency, the energy cost of remote data communication, and the cost inherent to Cloud infrastructure usage [27]. Through this, Dew computing optimizes the usage of mobile and IoT devices in two manners. First, it treats mobile devices as clients in the network infrastructure to offload their work to other devices located in the same network [28]. Second, Dew computing considers mobile and IoT devices as resources to increase the available computational power from an existing system. In this approach, one device can offload its work onto another available device in the network (including other mobile and IoT devices) [29,30].

We note that a network topology is needed in order to use mobile and IoT devices as resources in a local network [3]. The *Smart Cluster at the Edge (SCE)* is a network topology that is commonly used for that purpose in Dew computing. Figure 1 shows how the devices are organized in this type of network. This topology can be established wherever an access point and a group of mobile and IoT devices coexist. The topology's main feature is a central scheduler, which is primarily used to coordinate each task assignment among the network's available resources. This central scheduler can be any capable device in the network [31]. In this work, we address the problem of distributing jobs in a Dew environment under the assumption that the network topology is an SCE.

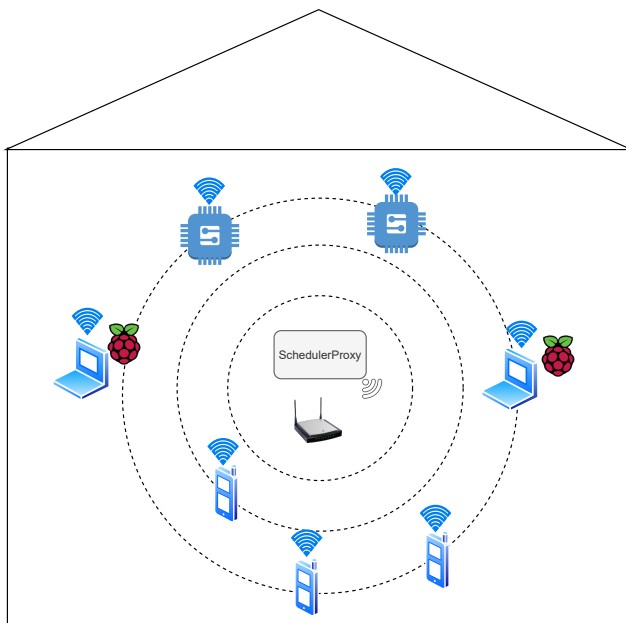

**Figure 1.** Example of a smart cluster at the Edge architecture. In this example, there are eight devices connected to the local network and communicating to a central scheduler. The devices include three smartphones, two raspberry pis, two PCs, and the device assigned as the scheduler.

## 3. Related Work

Distributing jobs in Dew environments *optimally* is a challenging combinatorial problem [32,33]. There are many factors that must be taken into account to properly assign jobs to devices. Those factors range from the device's CPU speed to how often jobs arrive at the scheduler. To deal with this complex problem, most previous works have proposed heuristic methods. A heuristic method is a fixed policy that considers some features from the SCE to distribute incoming jobs. In this paper, we explore an alternative approach, which consists of *learning* a policy to distribute jobs using RL. To provide a coherent view of the related work, we divided this discussion into two parts: We first discuss existing human-designed heuristic methods to distribute jobs in an SCE, and then, we review previous works at the intersection of RL with Cloud and Edge computing.

### 3.1. Human-Designed Heuristic Methods in SCEs

In the context of SCEs, different algorithms have been proposed to optimize the system's utility and job execution time, using the mobile devices remaining battery as a formal constraint of the resource allocation problem formulation [34–38]. These algorithms, however, assume complete and accurate information in terms of the energy spent and the execution time for every candidate node, making it challenging to apply in real-life scenarios.

Loke et al. [39] and Shah [40] addressed the previous limitation by proposing algorithms that do not rely on complete information. This approach seeks to exploit the nodes' proximity and cost-effectiveness of node-transferring capabilities. However, the focus of their study is to analyze the effect of nodes' mobility rather than balancing the load to efficiently utilize the battery and processing power of the available resources.

Hirsch et al. [31,32] presented other types of heuristics, which distribute jobs by considering the mobile devices' battery level and computing scores obtained from benchmarks. These methods outperformed traditional scheduling algorithms (such as Round Robin) but had one limitation: they only consider battery-dependent devices. This problem was recently addressed by Sanabria et al. [7]. Sanabria et al. [7] studied hybrid-mobile topologies, which combine both battery-dependent and non-battery-dependent devices and proposed heuristic methods that consider the device's battery level, computing score, and current work load in order to distribute jobs in Dew environments.

In contrast to those works, we propose to use RL to learn how to distribute jobs in a Dew environment. The main advantage of this approach is adaptability. As discussed by Sanabria et al. [7], no current heuristic seems to dominate across all Dew environments. This makes sense, since each Dew environment might have considerably different characteristics. Therefore, instead of trying to figure out (by ourselves) a clever way to correctly distribute jobs in any Dew environment, we propose to let an RL agent do that work for us (automatically).

### 3.2. RL Methods in Edge and Dew Computing

To the best of our knowledge, this is the first work exploring the use of RL for distributing jobs in Dew computing. However, RL has been used for task scheduling in closely related environments, such as in Cloud computing [13,41,42] and in Edge computing [15,43–49].

In the context of Cloud computing, task scheduling focuses on deciding which of the available resources (servers) should process an oncoming task. In this problem setup, previous works have shown that RL agents can reduce the execution time in distributed systems [41] and avoid overloading (and deadlocking) Cloud servers [13]. Furthermore, Cheng et al. [42] showed that deep RL can be helpful for scheduling tasks in large-scale Cloud Service Providers, surpassing traditional methods in terms of energy cost and reject rates.

As for Edge computing, the task-scheduling problem is similar. The Edge server receives tasks, and it has to decide whether to send those tasks to a nearby server or to the Cloud. Several works have explored how to distribute jobs in Edge computing using RL according to different performance metrics. These metrics include reducing computing times [43], energy consumption [49], latency [44,45], task failure rate [46], or a combination of the previous [15,47,48]. We note that in Edge computing, the action space is usually limited. For instance, sometimes the agent only has to decide whether to send (or not) a job to the Cloud [48]. In contrast, the action space in a Dew environment typically ranges from tens to hundreds (i.e., one action per device)—which makes the decision-making problem considerably harder in Dew computing.

There are three main differences between our work and the existing work on RL with Cloud and Edge computing. First, we address a different problem. Distributing jobs in a Dew environment has challenges that do not normally arise in Cloud computing or in Edge computing. For instance, in Dew computing, some of the devices might run out of

battery or users might start interacting with them. Taking these elements into account is a non-trivial task. Second, we explicitly study the generalization performance of the policies that the agent learned. This is a crucial step toward applying RL methods in real systems since, in practice, it is unlikely that the agent would encounter scenarios that it had previously seen during the training. Finally, we are using an RL agent that is better suited for generalization than the agents used in previous works [50,51].

## 4. Reinforcement Learning (RL)

RL agents learn optimal behaviors by interacting with an environment [9], where optimal behavior is defined with respect to a reward signal that the agent seeks to maximize.

Formally, the environment is modeled as a *Markov decision process (MDP)*. An MDP is a tuple $\mathcal{M} = S, A, r, p, \gamma, \mu$, where $S$ is a finite set of *states*, $A$ is a finite set of *actions*, $r : S \times A \times S \to \mathbb{R}$ is the *reward function*, $p(s_{t+1}|s_t, a_t)$ is the *transition probability distribution*, $\gamma \in (0, 1]$ is the *discount factor*, and $\mu$ is the *initial state distribution*, where $\mu(s_0)$ is the probability that the agent starts in state $s_0 \in S$. A subset of the states might also be labeled as *terminal states*.

At the beginning of an *episode*, the environment is set to an initial state $s_0$, which is sampled from $\mu$. Then, at time step $t$, the agent observes the current state $s_t \in S$ and executes an action $a_t \in A$. In response, the environment returns the next state $s_{t+1} \sim p(\cdot|s_t, a_t)$ and the immediate reward $r_{t+1} = r(s_t, a_t, s_{t+1})$. The process is then repeated from $s_{t+1}$ until potentially reaching a terminal state, when a new episode will begin.

The agent's goal is to collect as much reward from the environment as possible. To do so, the agent learns a *policy* $\pi(a|s)$, which is a probability distribution over the actions $a \in A$ given a state $s \in S$. Every policy induces a probability distribution over the future states and rewards that the agent will encounter if it selects actions according to such a policy. Therefore, we can rank different policies by how much reward they are expected to receive. Ideally, the agent will be able to improve its policy until finding an *optimal policy*, denoted by $\pi^*$, which is a policy that maximizes the expected reward received by the agent. Formally, optimal policies are defined as follows:

$$\pi^* = \text{argmax}_\pi \sum_{s \in S} \mu(s) \mathbb{E}_\pi \left[ \sum_{t=0}^{\infty} \gamma^t r_t \middle| s_0 = s \right] \tag{1}$$

Different methods have been proposed to learn optimal policies. Some classical examples include Q-learning [17] and SARSA [18]. Those approaches are known as tabular methods, since they use a large table to approximate $\pi^*$. Such a table includes one entry per state $s \in S$ and then, for each action $a \in A$, approximates the expected reward from executing action $a$ in state $s$. Note that the size of the table is equal to $S \times A$ and, thus, tabular methods are impractical when solving problems with really large state spaces—as is the case in Dew environments.

To solve problems with large (and potentially infinite) state spaces, deep RL methods use *deep neural networks* with parameters $\theta$ to model the policy $\pi_\theta(a|s)$. Neural networks are state-approximation techniques that, given a large enough network, can model any function [52]. In this case, the neural network $\pi_\theta$ receives as input a vector of real numbers (i.e., features) that represents the current state of the environment and outputs a probability distribution over the possible actions $a \in A$. We note that the output of the network depends on the parameters $\theta$, which are initially set to a random value. As a result, the initial policy $\pi_\theta(a|s)$ will tend to select all actions with equal probability. Then, the learning goal is to find $\theta^*$ such that $\pi_{\theta^*} \approx \pi^*$.

The key difference between deep RL methods lies in how they search for $\theta^*$. The first deep RL algorithm was *Deep Q-Network (DQN)* [53], which is a value-based method. After DQN, an actor-critic method, called, *Asynchronous Advantage Actor-Critic (A3C)* [22], was proposed. A3C learns much faster than DQN if we consider training time (although A3C is less sample efficient than DQN). Then, *Proximal Policy Optimization (PPO)* [19]

was proposed. PPO is an improved version of A3C that quickly became a state-of-the-art method. In fact, we note that A3C and PPO have shown strong generalization performance across different RL benchmarks [50,51,54]. However, that has not been the case for DQN [55]. For that reason, we experimented with A3C and PPO in this paper and left as future work the task of prototyping using other deep RL methods. Below, we provide further details about how PPO works, since PPO is the approach that achieved the best performance in our experiments.

PPO iteratively updates the parameters $\theta$ searching for a better policy (i.e., a policy that collects more reward). To do so, it first collects experiences by running $n$ agents in parallel for some fixed number of steps. Each agent collects experiences by sampling actions from the stochastic policy $\pi_\theta(a|s)$. Then, all those experiences are gathered together and become a training set that PPO uses to improve its current policy $\pi_\theta(a|s)$. This process is then repeated.

To update the parameters $\theta$, PPO uses gradient descent and a loss function that considers two main terms. The first term tries to predict how much expected return the current policy will obtain. That is, the network outputs one value, known as the *value function $v_\theta(s)$*, and updates $\theta$ towards making accurate predictions of how much reward the agent received form state $s$ in the training set. The second term updates $\theta$ toward increasing the reward that the policy $\pi_\theta(a|s)$ obtains. In short, PPO will increase the probability of selecting action $a$ from state $s$ if, whenever the agent selected $a$ in $s$ in the training set, the agent received more reward than expected. That is, if the difference between the reward that the agent received and $v_\theta(s)$ is positive, then PPO will update $\theta$ such that $\pi_\theta(a|s)$ increases (and decrease $\pi_\theta(a|s)$ otherwise). For more details on how and why PPO works, we refer the reader to the following papers [19,56].

## 5. Reinforcement Learning for Dew Computing

In this section, we discuss how we integrate Dew computing with RL. First, we formally define the problem of task scheduling in a Dew environment. Then, we describe how to solve such a problem using RL. Finally, we discuss the software architecture behind our proposed solution—which combines the Dew simulator EdgeDewSim [21] with the RL framework OpenAI Gym [20]. First, we briefly discuss the notation that we use in this section.

### 5.1. Notation

Below, we use the following notation. We use uppercase letters to refer to sets of elements and lowercase letters to refer to individual elements in those sets. For instance, we use $J$ to denote the set of possible jobs that arrive in the Dew environment and $j \in J$ to denote one particular job in $J$. In addition, $|J|$ denotes the number of elements in $J$. Elements in a set have different features. To refer to the value of a feature, we use $x.feature$. For instance, $j$.ops refers to the number of giga-operations of job $j \in J$.

### 5.2. Problem Definition: Task Scheduling in Dew Computing

We tackle the problem of distributing jobs in a Dew environment. The Dew environment consists of a set of devices connected to a local network. Some of these devices are *IoT devices* that have an unlimited power supply (e.g., Raspberry Pis and personal computers) and others are *mobile devices* with a limited battery (e.g., tablets and smartphones). In addition, one node in the local network is assigned as the *scheduler*. The scheduler's purpose is to receive jobs and distribute them among the devices, as shown in Figure 2.

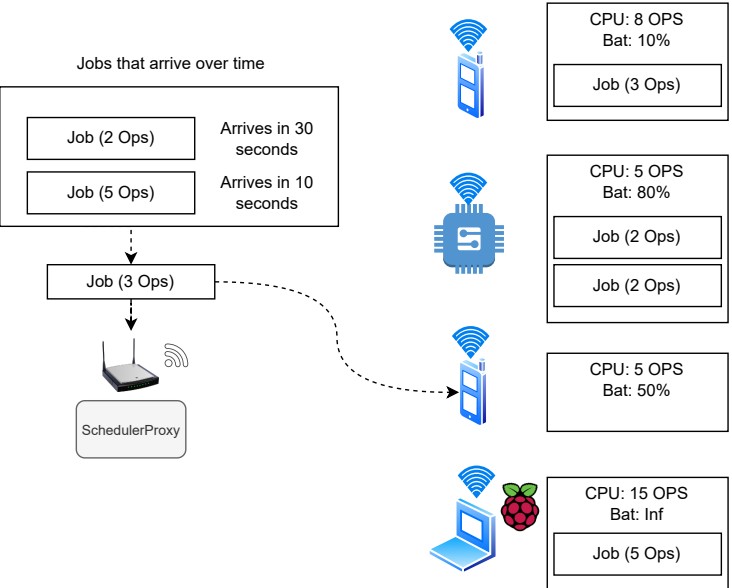

**Figure 2.** Overview of a scheduling problem in a Dew environment. Jobs arrive over time to the scheduler. Once a job arrives, the scheduler must assign that job to a device connected to the local network. In order to make such a decision, the scheduler has access to information about the job and the state of each device connected to the network. More details can be found in Section 5.3.

Once a job is assigned to a device, the time that it takes to complete the job (and how much battery it consumes) depends on the features of the job and the current state of the device. For instance, since some devices have limited battery, a job might not be completed if the job is assigned to a device with a low battery level. Users might also interact with the devices, and the local network might suffer from congestion issues. The scheduler must consider all these factors in order to distribute jobs effectively.

In more detail, assigning a job to a device adds that job to the device's job queue. Then, the device will keep running the jobs in its queue, one by one, until either completing all of them or running out of battery. Once a device runs out of battery, all the jobs in its current queue are discarded. If a job is assigned to a device that has no battery, that job is also discarded. In our experiments, if a job is discarded, it cannot be reassigned to a different device. This forces the scheduler to be extra careful when deciding to assign a job to a battery-dependent device.

To evaluate the effectiveness of a given job distribution, we use the *giga-instructions per second (GIPS)* that are completed in the Dew environment:

$$\text{GIPS}(t_0, t) = \frac{\sum_{j \in J_c} j.\text{ops}}{t - t_0},$$ (2)

where $J_c \subseteq J$ is the subset of jobs that were completed within the time interval $[t_0, t]$ and $j.\text{ops}$ is the number of giga-operations executed when processing job $j \in J_c$. Intuitively, GIPS is a measurement of how many operations are completed by a unit of time. The goal of the scheduler is then to distribute the jobs so that the GIPS are maximized in the Dew environment.

In this work, we assume that the Dew environment is fixed. That means that no device leaves or enters the local network during the period that we evaluate the system. We discuss how to extend our model to work in non-fixed Dew environments in Section 7.

### 5.3. Environment Definition: States, Actions, and Rewards

The first step to apply RL in Dew computing is to properly define the environment that the agent is going to interact with: that is, to define the states $S$ of the environment, the actions $A$ that the agent can perform, and the reward signal $r(s, a, s')$ that the agent will

optimize for. We might also have to define the transition probabilities $p(s'|s,a)$ if the agent does not interact with a real system (or with a predefined simulator). Whether the agent succeeds or fails at distributing jobs in a Dew environment partially depends on how we define those four elements: $S$, $A$, $r$, and $p$.

To define a Dew environment in terms of $S$, $A$, $r$, and $p$, our starting point was a state-of-the-art simulator called EdgeDewSim [7,21]. EdgeDewSim simulates real Dew environments with high accuracy. Among its many features, EdgeDewSim simulates the energy consumption of each device, the job executions, user behaviors, and basic network congestion. Thus, the transition probabilities $p$ in the environment are modeled by EdgeDewSim.

To define the state space $S$, we considered two key features of RL. First, most RL agents do not have memory. They learn a policy $\pi(a_t|s_t)$ that makes decisions purely based on the information available in the current state $s_t \in S$. Thus, $s_t$ must include all the relevant information that the agent needs in order to assign a job to the correct device. Second, we note that any information that is identical for all the states $s \in S$ is useless for the agent. The reason is that the agent has to discriminate whether action $a$ is a good action in state $s$. If a subset of information $x \subset s$ is identical for all states, then $x$ provides no discriminatory information to the agent.

We then define the state space as follows: $S = D \times J \times C$, where $(d, j, c) \in S$. Specifically, $d \in D$ contains information on the current state of each device. This information includes, for each device, its CPU usage percentage, its remaining battery, and its current job queue. Then, $j \in J$ provides information about the job that the scheduler must assign next. This information includes the job's ops, input size, and output size. Finally, $c \in C$ contains general statistics about the previously completed jobs. These statistics include the number of jobs that have arrived, the number of jobs that have been completed, the sum of the jobs' ops that have been completed, and the elapsed time since the beginning of the simulation (i.e., $t - t_0$). We note that $s \in S$ comprises all the necessary information to properly assign job $j$ to a device in a fixed Dew environment. If the environment is not fixed, then we might want to add some additional features, such as the number of instructions per second a device can execute, or how big its battery is. We discuss this further in Section 7.

Our definition of the action space $A$ is as expected. We define one possible action per device in the Dew environment. Then, whenever the agent executes action $a_i \in A$ given the current state $(d, j, c) \in S$, the job $j$ is assigned to the device associated with action $a_i$.

Finally, the reward function is equal to the GIPS executed in the Dew environment, as defined in Equation (2). Formally,

$$r(s, a, s') = \begin{cases} \frac{\sum_{j \in J_c} j.\text{ops}}{t - t_0} & \text{if } s' \text{ is terminal} \\ 0 & \text{otherwise} \end{cases} \tag{3}$$

Note that this reward function only rewards the agent in terminal states. That is, after the agent finishes distributing the whole sequence of jobs, the agent receives a reward that is equivalent to the GIPS executed in the Dew environment. As a result, any optimal policy $\pi^*(a|s)$ will, indeed, optimally distribute jobs according to our problem definition from Section 5.2.

As a brief summary of this section, Figure 3 illustrates the training loop and how a learning agent interacts with our proposed environment. The agent assigns the current job to a particular device using its actions. In response, the Dew environment returns the next state $s_t$ and a reward $r_t$. The state includes information about the current job to be assigned and the state of the devices. The reward $r_t$ will be zero unless the agent has just distributed the last job. If that is the case, $r_t$ will be equivalent to the GIPS. Since the agent tries to maximize the reward received from the environment, it will learn to distribute jobs in a way that improves the performance of the Dew environment.

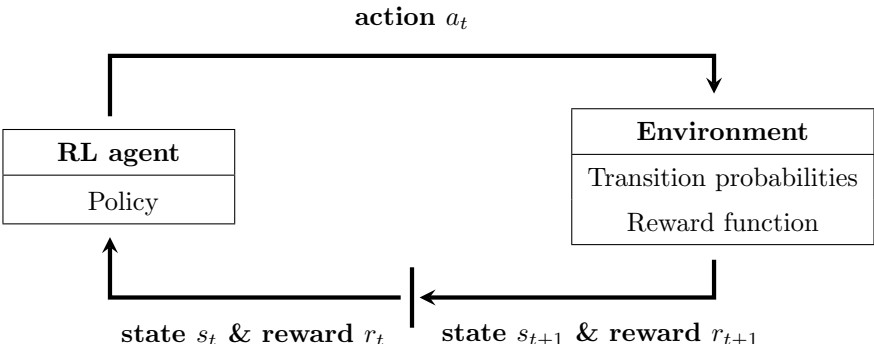

**Figure 3.** Training loop of the EdgeDewSim environment. The agent receives a job and the state of the devices. It then decides which device to assign the job to and sends that decision back to DewSim.

### 5.4. Implementation Details and Challenges

For the Dew environment simulator, we used the modified version of the DewSim simulator [21] proposed by Sanabria et al. [7]. This version can handle both battery-dependant and non-battery-dependant devices. To simplify the process of training and testing deep RL methods, we also developed an OpenAI Gym environment [20]. This allows us to quickly and easily test different RL agents using existing implementations made for the OpenAI Gym library.

Since EdgeDewSim is written in `java` and OpenAI Gym is written in `python`, we developed an interface that allowed us to control EdgeDewSim from OpenAI Gym. To do so, we added logic on top of EdgeDewSim to delegate the scheduling decision to an external agent, so that every time a job arrives, the simulator broadcasts the general state of the simulation and then listens for which device has to be selected, as shown in Figure 4. Then, we developed a new RL environment, following the guidelines from OpenAI Gym [20], that does three main things: (i) it establishes the connection with the simulator, (ii) it sends actions to the simulator, and (iii) it resets the environment to restart from an initial state. Further details about our implementation can be found in Appendix A.

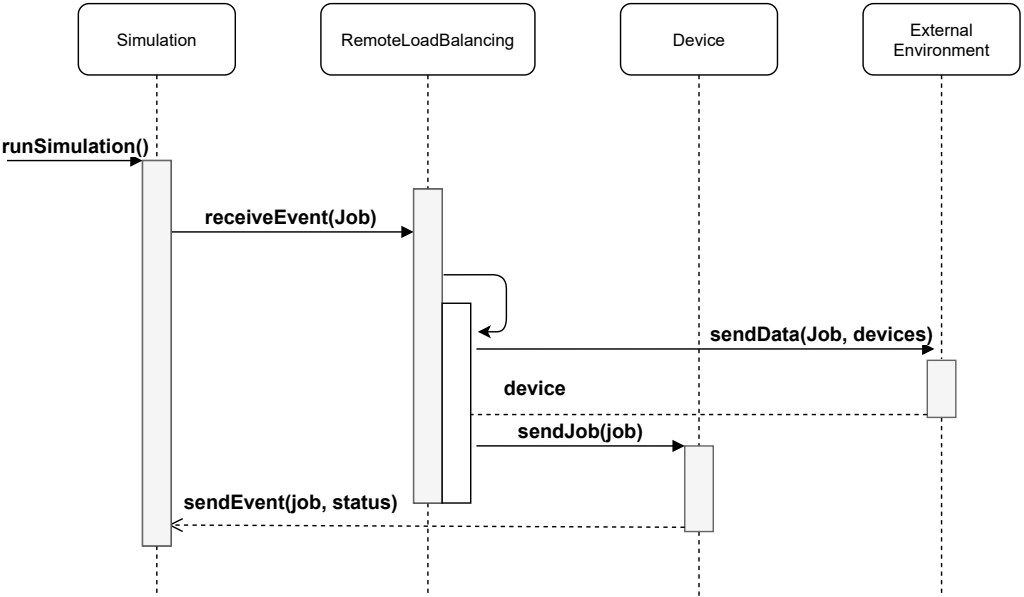

**Figure 4.** Sequence diagram of one job scheduling with an external environment.

In addition to implementing an efficient communication protocol between EdgeDewSim and OpenAI Gym, the main challenge toward using RL in Dew computing was to find a good set of hyperparameters for the agent. After careful fine tuning, we found a hyperparameter configuration that works well across all of our domains (as discussed below). However, we are not certain that those exact hyperparameters would work well in Dew environments that differ too much from the types of environments that we tested.

## 6. Experimentation

In this section, we provide an empirical evaluation of our method. The purpose of our experiments is to investigate the following research questions:

1. Can a deep RL agent learn to distribute jobs better than heuristic methods in a fixed Dew environment?
2. Can a deep RL agent learn a policy that generalizes well to unseen situations in a given Dew environment?

To answer these questions, we ran two sets of experiments. First, we show that deep RL is able to learn to distribute jobs better than human-designed heuristic methods in a *fixed job environment*. A fixed job environment is an environment where the agent is always presented with the exact same sequence of jobs and has to distribute them in a fixed Dew environment. We then discuss why such an approach is not expected to generalize well to unseen situations (i.e., other sequences of jobs) and show how to address that issue.

### 6.1. Baselines

We compare the performance of the RL agent with different heuristic methods. Each of these heuristic methods follows a fixed policy to distribute jobs in the Dew environment. They are the most common approaches for distributing jobs in Dew environments but, unfortunately, none of these algorithms dominates in every possible scenario [7]. In practice, depending on many factors, their performance can be strong or mediocre. In that regard, an RL agent has the advantage of being able to adapt its behavior to any particular situation.

Specifically, we compare the RL agent with the following heuristic methods:

- *Enhanced Simple Energy-Aware Scheduler (E-SEAS)* [32]: Before every allocation, the scheduler computes a score for each device which considers the node's current battery level, computation capabilities, and the number of jobs currently in its queue. Then, the device with the lowest score is selected.
- *Round Robin (RR)* [8]: The traditional RR algorithm gives one task to each device in a circular order until all tasks have been assigned.
- *Weighted Round Robin (W-RR)* [7]: Each device is assigned a weight proportional to its computing power. Then, when giving tasks to the devices in circular order, those with higher weights will be assigned more jobs at once.
- *Weighted Random (W-Rand)* [7]: Similarly to W-RR, the weight of each device is proportional to its computing power. The scheduler will then sample a random device when assigning a job, but those with higher weights will have a higher chance of being selected.
- *Batch Processing Algorithm (BPA)* [7]: In the same way as E-SEAS, the scheduler computes a score considering the node's current battery level, computation capabilities, and job load measured in how many operations are needed to finish each task. Then, the device with the lowest score is selected.

### 6.2. Fixed Job Experiments

We run our first set of experiments in two fixed job environments, which were proposed by Sanabria et al. [7]. The aim of these experiments is to answer our first research question. Below, we present the experimental setup, details on how we configured the deep RL agents, results, and a discussion about the limitation of this approach with respect to generalizing to new situations.

6.2.1. Experimental Setup

In this experiment, all the approaches have to distribute the same sequence of jobs over a predetermined set of devices. We used the same job sequence and device configuration as Sanabria et al. [7]. We note that Sanabria et al. [7] proposed such an experiment for the purpose of comparing heuristic methods in a complex Dew environment.

In more detail, the job sequence was generated using the standard methodology to create jobs in EdgeDewSim [21]. That is, each job is created by generating its input and output at random (between 1 and 500 MB). Then, the number of operations required to complete a job is computed as a function of the input data size. This function can be $n \cdot log(n)$, $n^2$, or $n^3$ (chosen at random for each task). The time of arrival to the scheduler was also sampled at random between 1 and 60,000 ms. Following this methodology, Sanabria et al. [7] proposed two datasets: one with 1500 jobs and another with 2500 jobs.

Regarding the devices, the Dew environments from these datasets use 70 devices with a constant source of energy and 50 devices with a limited battery supply. The non-battery-dependent devices consist of 35 Raspberry Pi 3 and 35 ODROID XU4. The battery-dependent devices consist of 10 Acer A100 tablets, 15 Samsung Galaxy Tab 2 tablets, and 25 LG L9 smartphones.

6.2.2. The Agent Configuration

In our experiments, we used the OpenAI baselines [57] implementation of PPO [19]. As explained in Section 4, PPO uses a deep neural network to model the policy $\pi_\theta(a|s)$ and the value function $v_\theta(s)$. In all our experiments, we used a feed-forward network with six hidden layers and 512 tanh units per layer to model $\pi_\theta(a|s)$ and $v_\theta(s)$. PPO then collects experience by running many agents in parallel for some number of steps. In our case, we let 96 agents collect experience until completing one episode. That is, they performed 1500 steps in the dataset with 1500 jobs and 2500 steps in the dataset with 2500 jobs. The resulting training set was used to update the parameters of the neural network. To do so, the training set was split into 96 minibatches, and the network was trained for 4 epochs using a learning rate of $10^{-5}$. The whole process is then repeated.

There are two additional hyperparameters that were key to making PPO work well in practice. Those are the *discount factor $\gamma$* and the *advantage estimation discounting factor $\lambda$*. Those hyperparameters control how much the agent discounts future rewards. Intuitively, in many problems, we want the agent to collect rewards as soon as possible. However, in a Dew environment, all the rewards are given at the end of the episode and, hence, discounting future rewards encourages the agent to repeat actions that received a large reward only because they were taken at the end of the episode. As a result, we recommend removing any form of discount by setting $\gamma = 1$ and $\lambda = 1$. The rest of the hyperparameters can be set to their default values from baselines [57].

6.2.3. Results

Figure 5 shows the results in the hybrid datasets with 1500 and 2500 jobs from [7]. The plots show the performance of each of the methods in GIPS as a function of the number of jobs that the agent has distributed so far. Note that the performance of the heuristic methods remains the same over time as they distribute jobs by following fixed policies. In contrast, the performance of PPO increases as the agent gains experience interacting with the Dew environment. In the case of PPO, the plot shows the average performance across 16 runs as well as one standard deviation in the violet shaded area and the maximum and minimum performance in light violet.

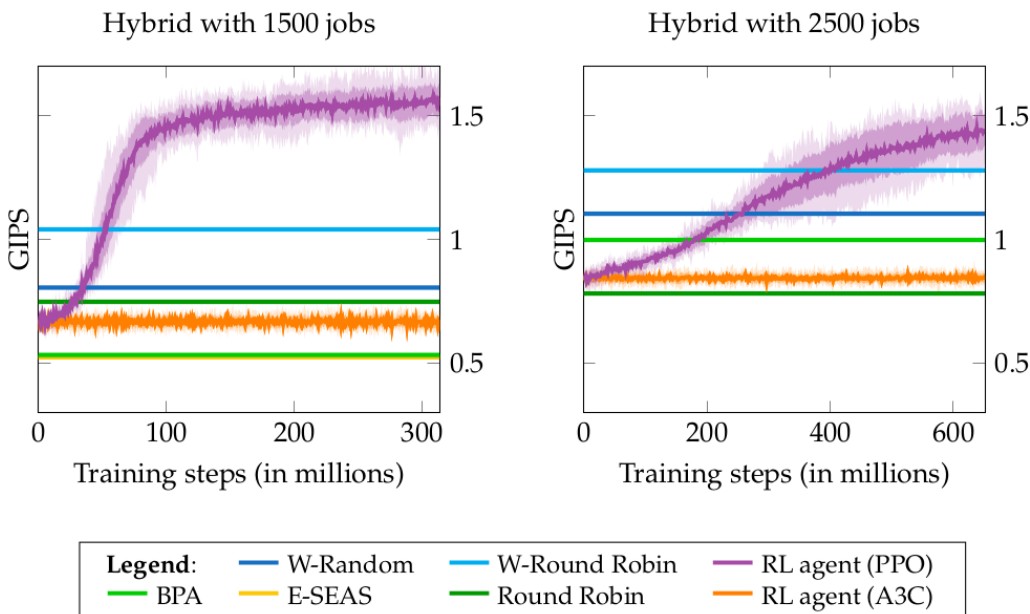

**Figure 5.** Training performance on the fixed job experiments. In both cases, the RL agent learned to distribute jobs better than existing state-of-the-art methods.

As Figure 5 shows, PPO eventually outperforms all the heuristics methods in both environments. In the 1500 jobs environment, PPO quickly improves and exceeds the performance of the baselines. With 2500 jobs, however, the agent requires around eight times more experience to outperform the baselines than with 1500 jobs. Part of the reason is that the baseline W-RR works particularly well with 2500 jobs. The other part is that the agent has to explore a larger space of possibilities when dealing with 2500 jobs than when distributing 1500 jobs. As a frame of reference, it took 5 days to train the agent with 1500 jobs and 10 days with 2500 jobs on an AMD Ryzen Threadripper 2990WX processor.

Interestingly, PPO outperforms all the baselines while being competitive at job completion. Recall that jobs can be lost in our environments if they are assigned to devices that are running out of battery. Even though we did not provide a reward to the agent for completing jobs, PPO learned to maximize GIPS while completing most of the jobs. As Table 1 shows, PPO completes more jobs than BPA, E-SEAS, and W-Random in both domains and is competitive with the Round Robin methods.

Finally, note that we also ran experiments using another RL agent, which is called A3C [22]. PPO could be considered an improved version of A3C. In particular, PPO has an extra constraint that A3C does not: PPO explicitly penalizes when the policy $\pi_\theta$ changes abruptly between two consecutive learning steps. As the results show, avoiding such large changes makes learning more stable and allows PPO to find good policies, while A3C does not. In these experiments, A3C is using the same hyperparameters as PPO. Since A3C is unable to find good policies in our experiments, we focus the discussion around PPO in the rest of the paper.

### 6.3. Discussion on Generalization

The previous results are encouraging. They show that RL agents can find ways to distribute jobs in Dew environments that are considerably better than distributing jobs using a heuristic method. However, it is still unclear whether the policy that the agent learned generalizes well: that is, whether that policy will work equally well on sequences of jobs that are different from the one used for training the agent. Indeed, it is possible that the agent is *overfitting* the sequence of jobs that was used for the training. The agent may have even memorized a sequence of actions that leads to high rewards in our specific training instance—which is a common behavior in deep RL [54,58].

We note that most *machine learning* models, including neural networks, assume that the training and test data are drawn from the same probability distribution in order to provide generalization guarantees [59]. In addition, the larger the training set is, the more likely it is that the model will generalize well. This behavior has also been observed in deep RL [50,54,55]. In short, the main conclusions from this body of work are two: (i) RL agents generalize better if they are presented with a diverse set of training instances, and (ii) the farther the testing instance is from the training distribution, the less likely it is that the agent will perform well (without been further trained).

To investigate this topic, we generated a set of 1000 job sequences (of different sizes) to test the generalization capabilities of the policy learned by the agent. Specifically, we generated six test sets. Each test set contains 1000 job sequences of a certain size, which were generated using the job generation procedure from EdgeDewSim [21] (as described in Section 6.2.1). The size of the sequence depends on the test set. We created test sets with sequences in the following sizes: 1500, 2500, 3500, 4500, 5500, and 6500 jobs. Note that the agent's performance on the test sets with 1500 and 2500 jobs will tell us whether the RL agent is generalizing to unseen instances that were sampled from the same distributions as the training instance. In contrast, the test sets with more than 2500 jobs allow us to study out-of-distribution generalization, where the longer the job sequences are, the harder it will be for the agent to generalize well.

Table 2 shows the performance of the RL agent (and the baselines) in each of the test sets. We tested two RL agents: PPO-1500 and PPO-2500. They are the same agent, but they were trained using the job sequence of 1500 jobs and 2500 jobs, respectively. Recall that we trained 16 agents per environment, so we are reporting the generalization performance of the agent that achieved the highest training performance among those 16 runs. The test performance was computed as the average GIPS across the 1000 testing instances. In addition, note that we are not further training PPO in any of the test environments. We are just running the learned policy of PPO-1500 and PPO-2500 and reporting their performance on each of the test sets.

**Table 1.** Percentage of job completed in the fixed job environments with 120 devices.

| Methods | BPA | E-SEAS | RR | W-Rand | W-RR | PPO |
|---|---|---|---|---|---|---|
| 1500 jobs | 0.87 | 0.85 | **0.89** | 0.84 | **0.89** | 0.88 |
| 2500 jobs | 0.76 | 0.74 | **0.80** | 0.74 | **0.80** | 0.78 |

**Table 2.** Upward generalization of policies that were learned over the fixed job environments with 120 devices and 1500 (or 2500) jobs measured in GIPS.

| Method | Train Set | | Upward Generalization | | | | | |
|---|---|---|---|---|---|---|---|---|
| Jobs: | 1500 j | 2500 j | 1500 j | 2500 j | 3500 j | 4500 j | 5500 j | 6500 j |
| BPA | 0.53 | 1.00 | 0.79 | 0.91 | 1.00 | 1.13 | **1.23** | **1.32** |
| E-SEAS | 0.52 | 0.85 | 0.74 | 0.82 | 0.86 | 0.94 | 1.02 | 1.09 |
| RR | 0.75 | 0.78 | 0.67 | 0.76 | 0.86 | 0.93 | 0.97 | 1.04 |
| W-Rand | 0.81 | 1.10 | 0.72 | 0.80 | 0.86 | 0.94 | 1.01 | 1.07 |
| W-RR | 1.04 | 1.28 | 0.68 | 0.77 | 0.84 | 0.90 | 0.97 | 1.01 |
| PPO-1500 | **1.73** | — | **1.35** | **1.46** | **1.35** | **1.25** | 1.18 | 1.14 |
| PPO-2500 | — | **1.60** | 1.06 | 1.23 | 1.23 | 1.21 | 1.19 | 1.17 |

Overall, Table 2 shows two expected and two unexpected results. The first expected result is that the performance of PPO decreases from the training set to the test set. For instance, the performance of PPO-1500 decreased from 1.73 to 1.35 in the test set with 1500 jobs. This is a symptom of overfitting, which is expected since we trained PPO-1500 using only one training instance. The second expected result is that the performance of PPO tends to decrease as we test on longer job sequences. In fact, BPA starts to dominate when the job sequences have 5500 or more jobs.

There are two (quite positive) unexpected results from Table 2. First, even though we trained using only one training instance, PPO still dominates in the test sets with 1500 to 4500 jobs. The second unexpected result is that the test performance of PPO-1500 with 2500 jobs was better than the test performance of PPO-2500 with 2500 jobs. As we discussed in Section 6.2.3, training PPO-2500 was considerably harder than training PPO-1500. However, the results from Table 2 suggest that it might be a better strategy to train the RL agents using short sequences of jobs and then gradually increase the length of the training sequences. The reason is that the agent is generalizing well to longer sequences and, hence, gradually increasing the size of the training sequence might allow us to train RL agents in Dew environments more effectively.

*6.4. Generalization Experiments*

One way of encouraging the agent to learn a policy that generalizes well is to show the agent a large collection of training instances that are sampled from the same distribution as the testing instances. We proposed a new environment to explore this idea. In this environment, the agent has to distribute a different sequence of jobs in every episode. As a result, the agent will have to learn a policy that works well for any possible job sequence in order to collect rewards. That includes performing well on the (unseen) testing instances.

Unfortunately, training an RL agent that can generalize to job sequences of different lengths and characteristics is more challenging. Indeed, previous works have shown that training RL agents that perform well across many training instances tends to require using considerably larger networks and more training experience [50,54] (i.e., more computational power). To reduce this computational burden, we decreased the number of devices in the Dew environment from 120 to 25 and trained the agent using job sequences of length of 300 to 500. Specifically, we used the following devices: seven Raspberry Pi 3, nine ODROID XU4, five Acer A100 tablets, two Samsung Galaxy Tab 2 tablets, and two LG L9 smartphones. The job sequences were generated using the methodology described in Section 6.2.1. Regarding hyperparameters, the RL agent used the same hyperparameters as those used for training the PPO-1500 model (see Section 6.2.2).

Figure 6 shows the average test performance over 1000 testing instances that were sampled from the same distribution as the training instances (i.e., job sequences of 300 to 500 jobs that were generated using EdgeDewSim). Note that in contrast to the plots from Section 6.2.3, Figure 6 is already showing the generalization performance of the RL agent. In particular, the plot shows the performance of all the baselines, the average performance of PPO across eight runs, and one standard deviation in the violet shaded area and the maximum and minimum performance across the eight runs in light violet. As the result shows, PPO is able to find a policy that generalizes well to unseen instances in addition to outperforming all the baselines.

To explore the agent's generalization capabilities further, Table 3 shows the test performance over test sets that were created in the same way as those from Section 6.3. The table reports the performance of all the baselines and the performance of the agent that achieved the highest training performance across the eight runs. As the table shows, there is no significant difference between the training and test performance of PPO in this case (0.54 vs. 0.55). Thus, there is no signal of overfitting. In addition, PPO outperforms all the baselines—even in sequences that are five times longer than the training sequences. That said, we can still see that the agent's performance decreases as the job sequences become longer. This is expected, since the agent optimized its behavior toward distributing up to 500 jobs during training.

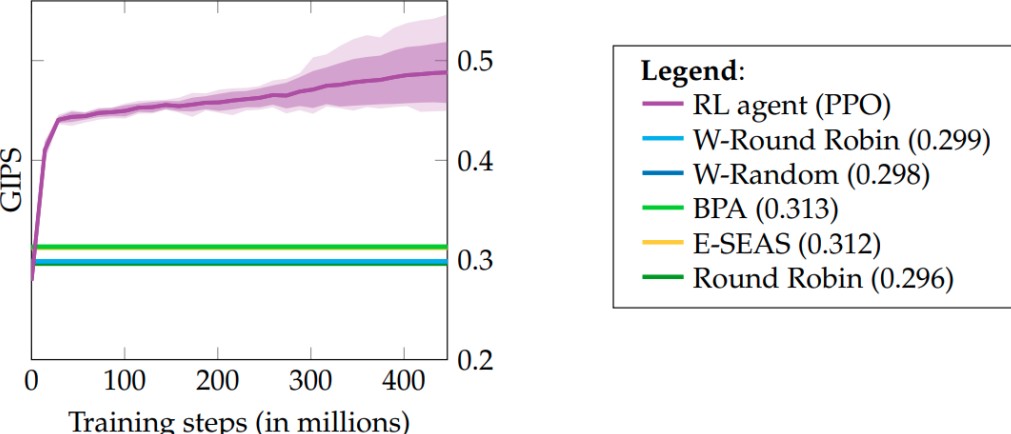

**Figure 6.** The plot shows the test performance (over 1000 testing instances) when the agent is trained using sequences of jobs sampled from the same distribution as the testing instances. Since some of the baselines' lines overlap, we also included their performance in the legend. Note that the RL agent quickly learned a policy that outperforms existing methods in the unseen testing instances.

**Table 3.** Upward generalization of policies that were trained using randomly generated job sequences in a Dew environment with 25 devices measured in GIPS.

| Method | Train Set | Upward Generalization | | | | |
|---|---|---|---|---|---|---|
| Jobs: | 300–500 j | 300–500 j | 1000 j | 1500 j | 2000 j | 2500 j |
| BPA | — | 0.31 | 0.34 | 0.38 | 0.40 | 0.42 |
| E-SEAS | — | 0.31 | 0.30 | 0.33 | 0.35 | 0.37 |
| RR | — | 0.30 | 0.30 | 0.31 | 0.33 | 0.35 |
| W-Rand | — | 0.30 | 0.30 | 0.32 | 0.34 | 0.37 |
| W-RR | — | 0.30 | 0.29 | 0.31 | 0.33 | 0.34 |
| PPO | 0.54 | **0.55** | **0.52** | **0.47** | **0.44** | **0.43** |

*6.5. Discussion*

Our experiments suggest positive answers to our two research questions. Deep RL did find policies that work better than heuristic methods and generalized well to unseen situations in our domains. In addition, we found two ways to achieve this behavior. One option is to train the agent using a single training instance. This method is computationally cheap and, as shown in Table 2, generalizes reasonably well. However, from a theoretical point of view, there are no guarantees that such an approach will generalize well in general. Another option is to train the agent using a different job sequence in every episode. This approach is computationally expensive but leads to policies that behave similarly in training and testing instances.

As for the limitations, the final policy learned by the agent is still a heuristic. This learned heuristic will perform well in situations that are similar to the training instances (and perform poorly in situations that differ too much from the training instances). Thus, the advantage of using RL is to allow users to learn an ad hoc heuristic for their particular Dew environment. The disadvantage is that the agent requires extensive training in order to learn such a heuristic. Searching for ways to decrease the cost of training the agent is a promising direction for future work.

**7. Conclusions and Future Work**

This work proposed to distribute jobs in Dew environments using RL. By following a trial-and-error strategy, the RL agent learned how to assign jobs to devices effectively—outperforming existing methods by up to 77% in GIPS performance. Perhaps our most

notable finding was that RL agents tend to learn policies that generalize well to unseen instances. This includes out-of-distribution examples, such as longer sequences of jobs than those seen during the training.

These are promising results. Beyond Dew computing, it is interesting to see a real-world application where Deep RL generalizes well from only one training instance. Investigating whether this is a common feature of large-scale scheduling problems (or only a peculiarity of Dew computing) is one of the many research directions that this work opens.

Another direction for future work is to study the performance of RL agents that are more *sample efficient* than PPO. As we said, PPO is one of the preferred methods for solving problems that require generalization. However, PPO is also known for being data-hungry—which is why PPO needed millions of interactions with the environment to become a state-of-the-art scheduler in our experiments. Now that we know that PPO works well in practice, it makes sense to try other RL methods that are more sample efficient, such as off-policy methods [60] and model-based methods [61].

In this work, we focused on scheduling jobs in a fixed Dew environment (i.e., an environment where the set of available devices is fixed). However, in some Dew environments, it is not uncommon that devices enter and leave the network over time [62]. This further complicates the process of assigning jobs to devices. We note that extending our method to handle such cases is relatively straightforward. The key is to show the RL agent situations in which a device leaves or enters the local network during the episode. That said, training the agent in such a setting will likely require more computational power.

Finally, it is worth mentioning that there is an important open question that we did not entirely address here. It is unclear how to evaluate the effectiveness of a scheduler in a Dew environment. There is no standard metric in the literature. Some metrics consider the GIPS [7], the number of jobs completed [31], the network latency [63], or the battery levels [64]. However, they all have advantages and disadvantages. In this work, we used GIPS because it pushes the agent to get the most out of the available resources. In order to maximize the GIPS, an optimal scheduler would keep all the available devices busy (if possible). Otherwise, the scheduler might improve its GIPS performance by assigning jobs to the IDLE devices. However, the disadvantage is that by doing so, the agent might decide to exploit the battery-dependent devices until their battery levels are close to zero. We leave as future work to study how RL agents behave under different performance metrics in Dew environments.

**Author Contributions:** Conceptualization, P.S., T.F.T., R.T.I. and A.N. methodology, P.S., T.F.T., R.T.I. and A.N. software, P.S., T.F.T. and R.T.I. validation, P.S., T.F.T. and R.T.I. formal analysis, P.S., T.F.T., and R.T.I. investigation, P.S., T.F.T., R.T.I. and A.N. resources, P.S., T.F.T., R.T.I. and A.N. data curation, P.S., T.F.T. and R.T.I. writing—original preparation, P.S., T.F.T., R.T.I. and A.N. writing—review and editing, P.S., T.F.T., R.T.I. and A.N. visualization, P.S. and T.F.T. supervision, R.T.I. and A.N. project administration, P.S., R.T.I. and A.N. funding acquisition, P.S., R.T.I. and A.N. All authors have read and agreed to the published version of the manuscript.

**Funding:** This work was funded by the National Agency for Research and Development (ANID)/ Scholarship Program/DOCTORADO NACIONAL/2020-21200979. We also gratefully acknowledge funding from the National Center for Artificial Intelligence CENIA FB210017, Basal ANID.

**Institutional Review Board Statement:** Not applicable.

**Informed Consent Statement:** Not applicable.

**Data Availability Statement:** Source code and datasets are available at: https://github.com/psanabriaUC/gym-EdgeDewSim, accessed on 8 July 2022.

**Conflicts of Interest:** The authors declare no conflict of interest.

## Appendix A. Simulator Details

In our previous work [7], we modified the original DewSim [21] to handle topologies based on hybrid networks; that is, we added the support of non-battery-dependent devices.

DewSim was chosen as a base simulator because it supports modeling different features present in an SCE. The main features of DewSim are the simulation of the arrival of tasks, completeness metrics, battery consumption, network activity derived from the transfer of input/output data of tasks, and status notifications of devices based on events. In addition, the Simulator allows modeling battery consumption and CPU usage in mobile devices thanks to its method based on profiles extracted from real devices, that is, traces that contain (not synthetic) information about the relationship between battery events and CPU usage. Battery depletion is simulated by executing events related to CPU usage, network usage, or screen activity. Figure A1 shows the logic of the Simulator to manage the battery status of the network's devices.

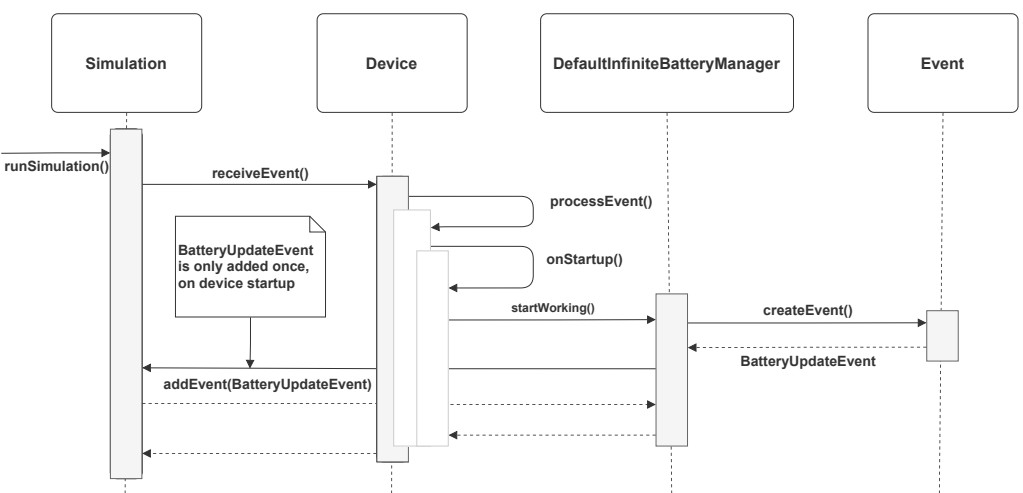

**Figure A1.** Sequence Diagram of DewSim battery management.

### Appendix A.1. Scheduling Logic

The Simulator manages the scheduling process using a base class called `SchedulerProxy`. This class has all the primary interaction and information required to handle the Events related to Job arrivals to the network. The scheduler has a method called `processEvent` that receives the Job, and then, based on its logic, using the data from the mobile devices, such as battery or CPU usage, and the data from the Job to process, it returns the selected device to perform the arrived Job. In Figure 3, we can see a class diagram showing how the different objects present in the Simulator are related.

In this work, to make it feasible for an RL agent to take the scheduling decision, we added a new Scheduler called `RemoteScheduler`. This Scheduler delegates the scheduling logic to an external entity by exposing a network interface. This interface exposes and sends the essential information, and then, it listens for which device has to be selected. Sent information includes job completion rate statistics, the list of devices with their corresponding assigned jobs, and energy status.

### Appendix A.2. Client/Server Mode

To integrate external scheduling algorithms made in other languages, we made new modifications to the Simulator. Firstly, we added a server mode that remotely helps control the simulation parameters and allows parallel simulation environments using a threaded model. After that, we created a `RemoteScheduler` class extending the base scheduler class. This `RemoteScheduler` does not do anything on its own and always waits for the instruction from an external entity. The flow begins with the server listening for connections, and when the client connects to the server, the server creates an entire Thread to attend the remote simulation. Next, the client sends the initial configuration using a defined communication protocol. Afterward, the server responds with an ACK response and begins the simulation until a scheduling decision needs to be made. When this scheduling decision arrives, the

Simulator passes the event to the `RemoteScheduler` class. It sends the current Simulation state, the current Job to assign, and the list of connected devices and waits for the client's response. The client then sends the ID of the assigned device to perform the Job and receives an ACK signal. This process is repeated until the simulation is finished or until the client sends a `reset` command. When the simulation ends, the server sends the client the last Simulation state and waits for a `restart` or `finish` command from the client. We show in Figure A2 the protocol and the logic followed by the Simulator and the client.

```
Client: Start Connection
Server: ACK, Create new thread and start a simulation instance
Client: Send string containing the configuration to use
        String length (n)          : 4 bytes Integer
        Null-terminated char array : N bytes char array
Server: Send simulation ID
        ID Least significant bytes : 8 bytes Long Integer
        ID Most significant bytes  : 8 bytes Long Integer
Client: ACK
(Simulation starts)
Server: Send Status: NEXT (2)
        Status:                    : 4 bytes Integer
Client: ACK
Server: Sends simulation metadata
        Device List size           : 4 bytes Integer
        Foreach device, send device metadata:
            If (firstTime)
                Send Full Data:
                    Device ID:        : 8 bytes Long Integer
                    MIPS              : 8 bytes Long Integer
                    Total Jobs        : 4 bytes Integer
                    Remaining Battery : 4 bytes Integer (Range 0-10000000)
                    CPU-Usage         : 8 bytes Double
                    Has Battery?      : 1 byte Boolean
                    AssignedJobs:     : 8 bytes Long
            Else
                Send Resumed Data:
                    Total Jobs        : 4 bytes Integer
                    Remaining Battery : 4 bytes Integer (Range 0-10000000)
                    CPU-Usage         : 8 bytes Double
                    AssignedJobs:     : 8 bytes Long
        Send Job Data:
            Job OPS                   : 8 bytes Long
            Job Input Size            : 4 bytes Integer
            Job Output Size           : 4 bytes Integer
        Send Statistics:
            Total Jobs                : 4 bytes Integer
            Completed Jobs            : 4 bytes Integer
            Elapsed time              : 8 bytes Long
            Successful Jobs OBS       : 4 bytes Float
Client: Send Assigned Device ID (or Reset command=0)
        Next Device               : 4 bytes Int
Server: ACK, Process with the next simulation action.
Repeats process Until Simulation finishes (or reset command is sent from the client)
Server: Sends END and final statistics Data:
        End Command (Value = 1)   : 4 bytes Integer
        Total Jobs                : 4 bytes Integer
        Completed Jobs            : 4 bytes Integer
        Elapsed time              : 8 bytes Long
        Successful Jobs OBS       : 4 bytes Float
Client: ACK, Send Next Command (RESET = 0, Finish = 1)
        Command                   : 4 bytes Integer
```

**Figure A2.** Client/Server communication protocol.

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
