# Peer review of "Solving Task Scheduling Problems in Dew Computing via Deep Reinforcement Learning"

_applsci, doi:10.3390/app12147137_

Round 1

Reviewer 1 Report

The authors Pablo Sanabria, Tomás Felipe Tapia, Rodrigo Toro Icarte, Andres Neyem presented a new solution for solving task scheduling problems in dew computing via deep reinforcement learning. No doubt the study is tangible but  some minor modifications are needed:

Must include some results in the abstract section.

Please separately add problem statement and contribution of the study in Introduction section.

Mention the challenges faced while implementation the proposed technique and how did you tackle those.

Mention if there are any trade-offs for implementation of this approach while we deviate from traditional practices of machine learning and implement deep reinforcement learning.

Some punctuation errors are observed. Please improve the punctuation in the manuscript.

Typo mistakes are there in the document.

In line 576-577 the authors write “But they all have advantages and disadvantages. In this work, we used GIPS because it pushes the agent to get the most out of the available resources.” Can you authors elaborate how GIPS make it possible to get maximum usage of resources.

In the Results section authors own insights required at the end of section.

Please elaborate conclusion by adding limitations of the proposed study and must be relevant to the proposed research.

Author Response

The authors Pablo Sanabria, Tomás Felipe Tapia, Rodrigo Toro Icarte, and Andres Neyem presented a new solution for solving task scheduling problems in dew computing via deep reinforcement learning. No doubt the study is tangible but  some minor modifications are needed:

Response: Thank you for the valuable feedback, we carefully reviewed all the questions and points included in the review and answered them point by point.

Point 1:  Must include some results in the abstract section.

Response 1: 

We added a summary of the results in the abstract section, the new Abstract is:

Due to mobile and IoT devices' ubiquity, and their ever-growing processing potential, Dew computing environments have been emerging topics for researchers. These environments allow resource-constrained devices to contribute computing power to others in a local network. One major challenge in these environments is task scheduling, that is, how to distribute jobs across devices available in the network. In this paper, we propose to distribute jobs in Dew environments using artificial intelligence (AI). Specifically, we show that an AI agent, known as Proximal Policy Optimization (PPO), can learn to distribute jobs in a simulated Dew environment better than existing methods -- even when tested over job sequences that are five times longer than the sequences used during the training. We found that using our technique, we can gain up to 77% in performance compared with using human-designed heuristics.

Point 2:  Please separately add the problem statement and contribution of the study in the Introduction section.

Response 2: 

We further separate the problem statement from our contributions in the introduction. Currently, the problem statement is described in lines 22-34, and our contributions are described in lines 51-77. 

Point 3:  Mention the challenges faced while implementing the proposed technique and how did you tackle those.

Response 3: 

In Section 5.4, we further discuss the challenges we faced while implementing our solution. In short, the main challenges were two: (1) implementing a communication protocol between DewSim and OpenAI GYM, and (2) finding a good set of hyper-parameters for the RL agent. We also discuss how we tackled those problems in this work.

Point 4:  Mention if there are any trade-offs for implementation of this approach while we deviate from traditional practices of machine learning and implement deep reinforcement learning.

Response 4: 

As discussed in the introduction (lines 69-77), there are trade-offs between distributing jobs using a learning-based approach (such as an RL agent) and using a human-design heuristic method. However, there are no clear trade-offs between using RL and other ML techniques. In fact, RL is the only traditional ML technique that applies to our problem because RL agents can autonomously learn how to distribute jobs in Dew environments. Other ML methods, such as supervised learning, require a labeled training set to work. In our problem, that dataset must include training pairs consisting of Dew environment problems and the optimal way to schedule all the jobs. Unfortunately, such a dataset does not exist and it would be too expensive to create one since we don't know an algorithm that can schedule jobs optimally in realistic Dew environments (nor exists human experts capable of accurately distinguishing optimal scheduling decisions). Thus, using RL seems the most sensible manner to tackle this problem using ML. Note that we largely discuss the advantages of using RL for learning to schedule jobs in Dew environments in the paper. 

Point 5:  Some punctuation errors are observed. Please improve the punctuation in the manuscript.

Response 5: 

We reviewed the whole manuscript and corrected the punctuation errors that we found.

Point 6:  Typo mistakes are there in the document.

Response 6: 

We reviewed the whole manuscript and corrected the typo mistakes that we found.

Point 7:  In lines 576-577 the authors write “But they all have advantages and disadvantages. In this work, we used GIPS because it pushes the agent to get the most out of the available resources.” Can you authors elaborate on how GIPS makes it possible to get maximum usage of resources?

Response 7: 

In order to maximize the GIPS, an optimal scheduler would keep all the available devices busy (if possible). Otherwise, the scheduler might improve its GIPS performance by assigning jobs to the IDLE devices. We clarified this in a footnote in Section 7.

Point 8:  In the Results, section authors' own insights are required at the end of the section.

Response 8: 

In the new version of our manuscript, we wrap up the experiment section with a discussion in Section 6.5. This section discusses how our results suggest affirmative answers to our research questions. That is, RL agents are a viable approach to scheduling jobs in Dew environments. We also provide our insights about our empirical results in sections 6.3 and 6.4.

Point 9:  Please elaborate conclusion by adding limitations of the proposed study and must be relevant to the proposed research.

Response 9: 

Thank you for this suggestion. We included a discussion about the limitations (and provided some directions for future work) in the conclusion section. See Section 7

Reviewer 2 Report

The reviewed paper proposes an implementation of an existing Deep Neural Networks-based Reinforcement Learning method for the problem of scheduling (distribution) of tasks in a dew computing system, where both power-supplied and batter-dependant devices are considered.

In my opinion, the topic of the paper is sound, relevant to modern-day challenges, and in the scope of the journal, including its targeted section and special issue.

The authors consider an approach that is novel in the context of Dew computing, however, more generally, the approach of employing RL methods for various problems (including scheduling) has become a common occurrence in recent years, lessening the novelty a bit.

The paper is generally well-written as well, though there are still some issues that the authors should address. The issues are listed before.

1. The SCE architecture shown in Fig. 1 is unclear. How it is actually defined? Is it just defined by the presence of battery-dependant devices and a central scheduler? Or is there more to it? In particular, Fig. 1 shows several inner and outer circles. Are those relevant to SCE itself or not?

2. The authors mentioned that the tasks can be lost due to the battery running out, at which point the task is effectively lost (if it was scheduled on such device already). Could the authors comment it the schedules obtained with their approach always resulted in all tasks being completed successfully?

3. The authors mentioned users being able to interact with their devices, effectively hampering the use of such devices for the dew environment. Was that included in the simulation in any way? Does the underlying simulator allow to simulate such events?

4. We know the tasks arrive over time (first minute if I remember right), but I am not sure if the problem is offline or online. Do we know how many tasks will arrive from the very start? Do we know everything about the job when it arrives?

5. Minor remark: as far as I understood the proposed RL algorithm does not guarantee an optimal solution in general. Thus, it is, formally at least, a heuristic as well. Perhaps it would be better to call other considered algorithms "constructive" or "(dispatch) rule-based" algorithms (assuming it is applicable to them).

6. The authors claim that no previous heuristic dominated over all instances. It is true that RL has the potential to generalize better (especially with diverse and well-chosen training sets), but it will most likely not dominate over all instances either, which stems from the No Free Lunch theorem. Perhaps that should be mentioned.

7. The problem definition (section 5.2) is very informal. The authors half-defined the concepts of j and J. I say "half" because we do not see what each job is composed of ("ops" appears informally out of the blue) and whether there are any restrictions on the data (are "base" processing times in seconds? Milliseconds? Are they just integers or allowed to be a real/float number)? The set of machines/devices is not defined. The constraints are not well-defined. We do not know if a device can process only one job at a time. We do not know if all jobs have to successfully completed (forceful and formal problem constraint) or we just hope so but do not enforce it. In short: please formalize the mathematical model.

8. Perhaps showing a very small exemplary instance (3-5 jobs, 2-3 devices) would be helpful.

9. I was a bit confused about how many instances were used in training. Perhaps this should be made more clear.

10. Why is A3C "constant" in Fig. 4? Does it not improve/learn at all? Why would that be?

11. What values are shown in Table 1? Is it rewards? Goal function? Giga-ops over time unit? Please clarify.

12. Fig. 5 seems to miss several plotlines. Where are they?

13. The conclusions are too short. Perhaps some numerical results from the paper should be summarized? The same goes for the abstract: it seems too plain as if the authors had no results to put there, which is not true.

14. Last, but perhaps one of the most important issues: due to the black-box nature of most Machine Learning approaches, it is almost mandatory to provide source code for the research to let others verify the results. Please include the code necessary to reproduce the result (this can be a link to some site as arXiv or a Github repository).

There are some editorial issues:

-- "written java", "written python"

-- In line 245 "A" should be changed to Appendix. A

-- The list of abbreviations seems vastly unnecessary.

Author Response

The reviewed paper proposes an implementation of an existing Deep Neural Networks-based Reinforcement Learning method for the problem of scheduling (distribution) of tasks in a dew computing system, where both power-supplied and batter-dependant devices are considered.

In my opinion, the topic of the paper is sound, relevant to modern-day challenges, and in the scope of the journal, including its targeted section and special issue.

The authors consider an approach that is novel in the context of Dew computing, however, more generally, the approach of employing RL methods for various problems (including scheduling) has become a common occurrence in recent years, lessening the novelty a bit.

The paper is generally well-written as well, though there are still some issues that the authors should address. The issues are listed before.

Response

We appreciate you for your precious time in reviewing our paper and providing valuable comments. It was your valuable and insightful comments that led to possible improvements in the current version. The authors have carefully considered the comments and tried our best to address every one of them. We hope the manuscript after careful revisions meet your high standards. The authors welcome further constructive comments if any. Below we provide the point-by-point responses. We marked all the modifications using the changes LaTeX package to mark the differences in the paper source code, and in the manuscript, all modifications have been highlighted in blue.

Point 1. The SCE architecture shown in Fig. 1 is unclear. How it is actually defined? Is it just defined by the presence of battery-dependant devices and a central scheduler? Or is there more to it? In particular, Fig. 1 shows several inner and outer circles. Are those relevant to SCE itself or not?

Response 1: 

The SCE architecture is defined by a scheduler and a set of devices connected to a local network. We included the circles to highlight that all the devices are connected to the same network. This was clarified in the caption of Fig 1.

Point 2. The authors mentioned that the tasks can be lost due to the battery running out, at which point the task is effectively lost (if it was scheduled on such device already). Could the authors comment it the schedules obtained with their approach always resulted in all tasks being completed successfully?

Response 2: 

Not necessarily. We are not rewarding the agent for completing jobs, so the agent would learn to complete jobs only to the extent that it aligns well with maximizing the GIPS. That said, the agent is still completing most of the jobs. The 1500 job experiment completes around 88% of the jobs whereas the best performing baseline (W-RR) completes 89% of the jobs. In the new version of our manuscript, we report job-completing results in Table 1 and discuss them in Section 6.2.3. 

Point 3. The authors mentioned users being able to interact with their devices, effectively hampering the use of such devices for the dew environment. Was that included in the simulation in any way? Does the underlying simulator allow to simulate such events?

Response 3: 

Yes, the simulator uses historical data of CPU usage to simulate users interacting with their mobile devices. Thus, the RL agent must learn to take those events into account in order to effectively distribute jobs in Dew environments. We discuss this in Appendix A.

Point 4. We know the tasks arrive over time (first minute if I remember right), but I am not sure if the problem is offline or online. Do we know how many tasks will arrive from the very start? Do we know everything about the job when it arrives?

Response 4: 

Our approach is online in the sense that the agent does not know how many jobs are going to arrive. However, once a job arrives, the agent knows its giga-operations (ops), input size, and output size (as described in Section 5.3). We note that our method is fairly general. In principle, the RL agent can learn to exploit any available information in order to assign jobs to devices. However, the quality of the resulting scheduling will depend on the quality of that information. Our results show that, if the agent has access to the job's ops, input size, and output size, it can learn to distribute jobs better than existing methods. Note that all our baselines (except for RR) use the jobs ops, input size, and output size to schedule jobs in one or another way. Thus, our comparison is fair in that regard.

Point 5. Minor remark: as far as I understood the proposed RL algorithm does not guarantee an optimal solution in general. Thus, it is, formally at least, a heuristic as well. Perhaps it would be better to call other considered algorithms "constructive" or "(dispatch) rule-based" algorithms (assuming it is applicable to them).

Response 5: 

This is a good point. The key difference between our method and the baselines is that our approach learns to distribute jobs whereas the baselines follow a fixed policy designed by a human. After some discussion, we settled on naming our baselines as "human-design heuristics" in the paper. We believe that such a name is better than rule-based or constructive heuristics because the RL agent might also learn a policy that could be considered rule-based (or constructive).

Point 6. The authors claim that no previous heuristic dominated over all instances. It is true that RL has the potential to generalize better (especially with diverse and well-chosen training sets), but it will most likely not dominate over all instances either, which stems from the No Free Lunch theorem. Perhaps that should be mentioned.

Response 6: 

Indeed, RL agents learn a heuristic that performs well in situations that are similar to the training instances and poorly in situations that differ too much from the training data. We added a discussion about this limitation in Section 6.5.

Point 7. The problem definition (section 5.2) is very informal. The authors half-defined the concepts of j and J. I say "half" because we do not see what each job is composed of ("ops" appears informally out of the blue) and whether there are any restrictions on the data (are "base" processing times in seconds? Milliseconds? Are they just integers or allowed to be a real/float number)? The set of machines/devices is not defined. The constraints are not well-defined. We do not know if a device can process only one job at a time. We do not know if all jobs have to successfully completed (forceful and formal problem constraint) or we just hope so but do not enforce it. In short: please formalize the mathematical model.

Response 7: 

We partially disagree with the reviewer on this particular point. We are solving a problem for which there is no mathematical model. If we could model this problem using mathematical programming, we would have solved it using MIP or CP, and the solution we find would have worked only under the assumptions that such a model makes.

RL is different from mathematical programming. RL does not require a mathematical definition of the problem in order to solve it. RL only requires access to an environment that receives actions and returns states and rewards. For that reason, our problem definition only discusses the minimal elements that are required in order to apply our method (Section 5.2). That is, there has to be a scheduler that receives jobs and distributes them among a fixed set of devices. And the goal of the scheduler is to maximize the GIPS performed in the environment. Other details, such as the features that are available for each job (and device) or how a device processes a job are not relevant. Our method will still apply regardless of how each of those components works.

That said, we agree that it is important to know those details in order to fully understand our experimental results. Therefore, we explain the features we are using in Section 5.3 and how DewSim works in sections 5.3, 5.4, and Appendix A. In a way, the closest to a mathematical model of our problem would be the code of DewSim. That code is publicly available and described in two papers [7,21]. For those reasons, we consider that the level of details at which we are formalizing our problem is appropriate.

Below, we provide answers to your specific questions:

  1. Are there any restrictions on the data (are "base" processing times in seconds? Milliseconds? Are they just integers or allowed to be a real/float number))?

There are no restrictions on the data. Recall that we are feeding this information to a neural network. And neural networks can handle discrete and continuous features. In addition, the scale of those features (whether they are seconds or milliseconds) is not relevant because they are normalized when computing the gradients via batch normalization.

  1. We do not know if a device can process only one job at a time.

In our experiments, each device processes one job at a time. We explain this in Section 5.2: "the device will keep running the jobs in its queue, one by one until either completing all of them or running out of battery."

  1. We do not know if all jobs have to successfully completed (forceful and formal problem constraint) or we just hope so but do not enforce it.

There is no constraint enforcing the completion of jobs. RL agents learn a policy that maximizes the expected return (see equation 1). That optimization problem is unconstrained so there is nothing enforcing that the agent completes all the jobs. We added a discussion about job completion in Section 6.2.3.

Point 8. Perhaps showing a very small exemplary instance (3-5 jobs, 2-3 devices) would be helpful.

Response 8: 

Thank you for this suggestion. We included a small diagram (Figure 2) to better illustrate the scheduling problem in a Dew environment. 

Point 9. I was a bit confused about how many instances were used in training. Perhaps this should be made more clear.

Response 9: 

The fixed-job experiments use only one training instance (see section 6.2 for more details).

In the generalization experiments from Section 6.4, we trained on a different instance (i.e., sequence of jobs) in every episode. Each episode consisted of 300 to 500 jobs. Thus, we can estimate the number of training instances from Figure 6 by dividing the number of training steps into 400. Note that PPO outperformed the baselines after 14 million training steps in Figure 6. That is, after seeing approximately 35,000 training instances. 

Point 10. Why is A3C "constant" in Fig. 4? Does it not improve/learn at all? Why would that be?

Response 10: 

Intuitively, the problem with A3C is that it tends to be too optimistic under uncertainty. Imagine that the agent has never tried an action before in a given state. Should it assume that the action is good or bad? A3C tends to think that the action is good and quickly moves its policy towards trying that action. But this greedy behavior makes learning unstable. Sometimes increasing the likelihood of selecting unknown actions improves the scheduling performance. And sometimes it causes the opposite. For that reason, A3C keeps jumping (locally) between good and bad policies. In contrast, PPO is a trust-region method that prefers to change its policy towards known regions only. This property allows PPO to improve its performance steadily.

The theoretical justification for this behavior can be found in Schulman et al. [56].

Point 11. What values are shown in Table 1? Is it rewards? Goal function? Giga-ops over time unit? Please clarify.

Response 11: 

All the tables show the performance of the different methods in GIPS. We clarified this in their captions.

Point 12. Fig. 5 seems to miss several plotlines. Where are they?

Response 12: 

Our plotline was complete, but it was hard to see the results because some of the lines overlap. We now discuss this overlap in the caption and include the performance of each baseline in the legend.

Point 13. The conclusions are too short. Perhaps some numerical results from the paper should be summarized? The same goes for the abstract: it seems too plain as if the authors had no results to put there, which is not true.

Response 13: 

Thank you for your suggestion. We included a summary of our results in the abstract and conclusion. We also added a discussion about the limitations of our method and directions for future work to the conclusion. 

Point 14. Last, but perhaps one of the most important issues: due to the black-box nature of most Machine Learning approaches, it is almost mandatory to provide source code for the research to let others verify the results. Please include the code necessary to reproduce the result (this can be a link to some site such as arXiv or a Github repository).

Response 14: We published in a Github repository the Simulator and the RL Agent, the link is “https://github.com/psanabriaUC/gym-EdgeDewSim”. We also updated the link in the article. That change is located in line 623

Point 15: There are some editorial issues:

-- "written java", "written python"

-- In line 245 "A" should be changed to Appendix. A

-- The list of abbreviations seems vastly unnecessary.

Response 15: We reviewed those edition issues and removed the abbreviations list.